# MambaTree: Tree Topology
# is All You Need in State Space Model

**Yicheng Xiao**[1][†][*],     **Lin Song**[2,3][✉][*],
**Shaoli Huang**[3], **Jiangshan Wang**[1], **Siyu Song**[4], **Yixiao Ge**[2,3], **Xiu Li**[1][✉], **Ying Shan**[2,3]

[1]Tsinghua Shenzhen International Graduate School, Tsinghua University
[2]ARC Lab, Tencent PCG   [3]Tencent AI Lab   [4]South China Normal University
xiaoyc23@mails.tsinghua.edu.cn   ronnysong@tencent.com

## Abstract

The state space models, employing recursively propagated features, demonstrate strong representation capabilities comparable to Transformer models and superior efficiency. However, constrained by the inherent geometric constraints of sequences, it still falls short in modeling long-range dependencies. To address this issue, we propose the MambaTree network, which first dynamically generates a tree topology based on spatial relationships and input features. Then, feature propagation is performed based on this graph, thereby breaking the original sequence constraints to achieve stronger representation capabilities. Additionally, we introduce a linear complexity dynamic programming algorithm to enhance long-range interactions without increasing computational cost. MambaTree is a versatile multimodal framework that can be applied to both visual and textual tasks. Extensive experiments demonstrate that our method significantly outperforms existing structured state space models on image classification, object detection and segmentation. Besides, by fine-tuning large language models, our approach achieves consistent improvements in multiple textual tasks at minor training cost. Code is available at https://github.com/EasonXiao-888/GrootVL.

## 1 Introduction

Mainstream fundamental models are primarily based on CNN [30, 62, 44, 32, 13] and Transformer architectures [15, 43, 42, 59, 14], which dominate in visual and language tasks. However, the small receptive field of CNNs and the high complexity of Transformers make it challenging to strike a good balance between effectiveness and efficiency. The state space models (SSMs) [22, 24, 52] attempt to disrupt this impasse, which model sequences in a recurrent form. Different from the previous recurrent neural networks [31, 7], these approaches draw inspiration from control systems, leveraging structural parameter initialization to attain stable optimization and superior computing performance. Nevertheless, it remains susceptible to the intrinsic flaw shared by recurrent neural networks, $i.e.$, a deficiency in capturing long-range dependencies.

Recently, an improved selection mechanism known as Mamba [19] is proposed to mitigate the challenges of SSMs. This approach introduces weight modulation during the propagation process, which substantially enlarges the effective receptive field and achieves impressive performance in NLP tasks. Besides, numerous studies aim to extend Mamba into computer vision, by employing various pre-defined strategies to map 2D image features into 1D sequences. ViM [77] and VMamba [41] utilize a multi-directional raster-scanning strategy, while LocalMamba [34] further confines its

---

*Equal contribution. † Work done during an internship at Tencent. ✉ Corresponding author.

38th Conference on Neural Information Processing Systems (NeurIPS 2024).

propagation range within a local window. They have successfully adapted Mamba to image inputs. Nevertheless, as shown in Fig. 1(a), both raster-scanning and local-scanning strategies introduce spatial discontinuities between adjacent pixels, and feature transformations in Mamba rely on the feature relationships, thereby impeding the effective information flow in a sequence. Additionally, PlainMamba [69] introduces a continuous scanning strategy, aiming to alleviate this issue by simply adjusting the propagation direction at discontinuous positions. However, all these methods rely on fixed propagation trajectories, which ignore the inherent spatial structure and cannot dynamically adjust the topology based on input. Therefore, this paper endeavors to explore a new perspective: *introducing an input-aware topological network for feature propagation in state space models.*

To achieve it, we develop a tree state space model and propose a new framework, termed MambaTree, which adaptively generates a tree topology based on the input feature and then performs feature propagation on it. Specifically, two sub-networks, MambaTreeV and MambaTreeL, are designed for visual and language tasks respectively, which are illustrated in Fig. 1(b) and Fig. 1(d). For visual tasks, motivated by [71, 54], we first utilize the dissimilarity between adjacent features to construct a minimum spanning tree on a four-connected planner graph. This process can adaptively encode the spatial and semantic information into a tree graph [71, 54]. Then, we iteratively traverse each pixel, considering it as the root vertex, and aggregate the features of other pixels using the state transition function of Mamba. Intuitively, this operation requires two levels of traversal across the entire pixel set, resulting in an unacceptable quadratic complexity relative to the number of pixels. However, given that the tree graph is acyclic, we propose a dynamic programming algorithm to achieve linear complexity propagation. With such an input-aware tree topology, our approach enables more effective long-range interactions while maintaining consistent linear complexity with Mamba. Furthermore, our method can also be applied to language tasks by constructing a tree topology based on the dissimilarity between token features, which overcomes the geometrical constraints of the text sequence. Using a similar aggregation process as MambaTreeV, MambaTreeL can significantly enhance the language representation of a pre-trained Large Language Model [19].

We conduct extensive experiments to validate the effectiveness of MambaTreeV on multiple visual benchmarks, *i.e.* image classification on ImageNet [12], object detection and instance segmentation on MSCOCO [39] as well as semantic segmentation on ADE20K [75]. Results show that our method notably outperforms existing SSM-based methods for all benchmarks and achieves competitive performance with CNN and Transformer-based approaches. Moreover, with LoRA finetuning [33], MambaTreeL demonstrates consistent improvements for a pre-trained large language model at minor training cost.

## 2 Related Work

### 2.1 Conventional Vision Foundation Models

The evolution of deep neural networks has been a significant catalyst in machine vision perception. CNN-based models [30, 51, 35, 25, 61, 72, 38, 55, 73] firstly emerge as pivotal landmarks, with ResNet [30] notably standing out for its inventive residual connection module, garnering widespread adoption across diverse domains of visual recognition. Furthermore, more efficient convolution operations are formulated, such as depth-wise convolutions introduced by MobileNet [32], paving the way for lightweight models. Additionally, deformable convolution [10] has been proposed to enhance the receptive field. Subsequently, ViT [15] has significantly improved the vision recognition paradigm. It reformulates the architecture design and training mechanism by combining transformer architecture in natural language processing, aiming to improve computational efficiency and broaden the scope of applications. After research discourse is centred on hierarchical ViTs [43, 42, 11, 63, 14, 56, 5] which design networks by decreasing feature resolution across the backbone gradually. Furthermore, recent research built on CNN serves to re-emphasize the capabilities of convolutional networks. For example, InternImage [62] presents a large model based on deformable CNN, while UniRepLKNet [13] exhibits significant performance through large kernel convolution.

### 2.2 Explorations about State Space Models

State space models (SSMs) have emerged as a novel class of models within the deep learning paradigm, showing significant potential for sequence transforming [23, 22, 52]. These methods have attracted significant attention due to their linear scalability with sequence length. The early method,

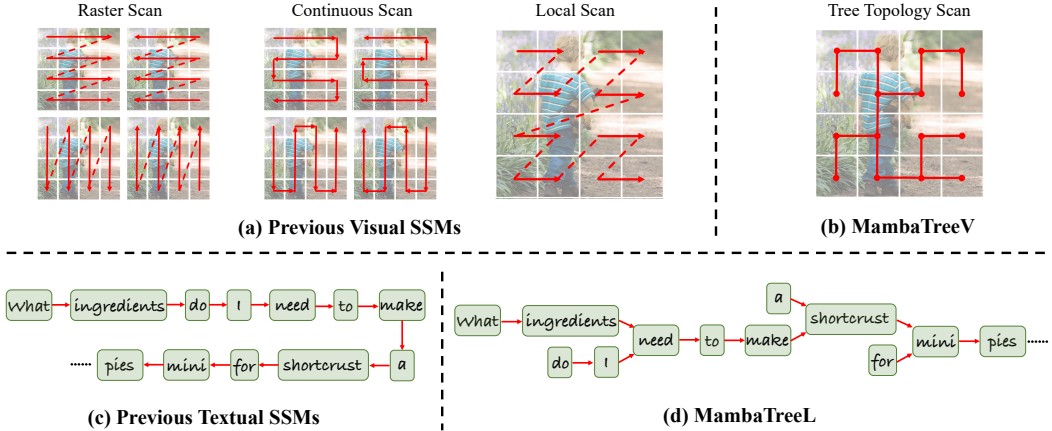

**(a) Previous Visual SSMs**

**(b) MambaTreeV**

**(c) Previous Textual SSMs**

**(d) MambaTreeL**

Figure 1: **Comparison of different propagation strategies for multi-modal tasks.** For visual tasks, the previous strategies (a) are based on fixed patterns, while our method can adaptively generate the propagation topology according to input features. For textual tasks, compared to previous methods (c), our approach (d) can break the inherent constraints of text sequences, facilitating the effective transmission of long-range information.

LSSL [23], draws inspiration from continuous state space models in control systems and attempts to address the long-range dependency problem through a combination with HIPPO [20] initialization. S4 [22] proposes to normalize the parameters into a diagonal matrix, prompting a subsequent series of research on structured SSMs [24, 21, 26, 19]. Recently, the Selective State Space Model [19], known as Mamba, strikes a balance between effectiveness and efficiency through the design of an input-dependent parameter initialization strategy, which has emerged as a formidable competitor to both transformer and CNN structures. In addition to showcasing superior outcomes in sequence modeling, Mamba has been seamlessly incorporated into the visual domain [77, 41, 34, 69, 68]. These studies often rely on handcrafted fixed scanning mechanisms to mitigate the execution bias of the selective state space model on 2D non-causal images. However, such simplistic approaches cannot effectively capture spatial relationships in an input-dependent paradigm. To address this limitation, we propose an effective framework MambaTree in this work to enhance long-range modeling for both vision and language tasks by introducing an input-aware tree-based topological structure.

## 3 Method

In this section, we first revisit the selective state space model [19] and then elaborate on our input-aware topology scanning algorithm for state space modeling. With this superior algorithm, we develop a tree SSM and propose a novel framework called MambaTree, which consists of two sub-networks: MambaTreeV for visual tasks and MambaTreeL for fine-tuning a pre-trained language model [19].

### 3.1 Revisiting Selective State Space Model

State Space Models (SSMs) are commonly regarded as continuous linear time-invariant systems [64] that map input stimulation $x(t) \in \mathbb{R}^{1 \times D}$ to output signal $y(t) \in \mathbb{R}^{1 \times D}$ through a state vector $h(t) \in \mathbb{R}^{1 \times N}$, where $t$, $D$ and $N$ indicate the time step, channel number of the signal and state size, respectively. These models can be formulated as the following linear ordinary differential equations:

$$h'(t) = \mathbf{A}h(t) + \mathbf{B}x(t), \quad y(t) = \mathbf{C}h(t) + \mathbf{D}x(t), \tag{1}$$

where $\mathbf{A} \in \mathbb{R}^{N \times N}$, $\mathbf{B} \in \mathbb{R}^{N \times D}$, $\mathbf{C} \in \mathbb{R}^{N \times D}$ and feedthrough coefficient $\mathbf{D} \in \mathbb{R}^{D}$.

**Discretization.** Although SSM serves as a powerful tool in systems and control engineering, its time-continuous nature poses challenges for integration into deep learning architectures. To alleviate this issue, most methods utilize the zero-order hold rule [19] to discretize the continuous system described by Eq. (1) and convert continuous variables ($\mathbf{A}$, $\mathbf{B}$, $\mathbf{C}$, $\mathbf{D}$) into corresponding discrete

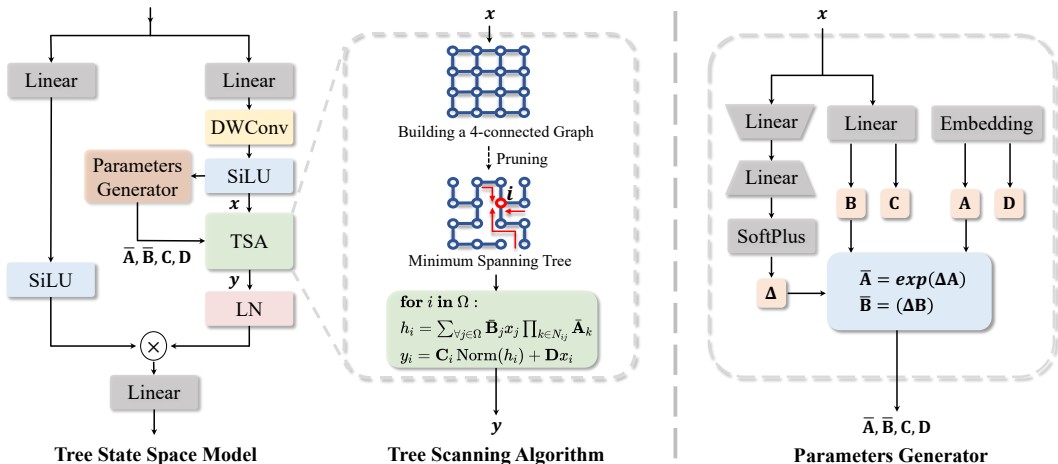

| Tree State Space Model | Tree Scanning Algorithm | Parameters Generator |

Figure 2: **Illustration of Tree State Space Model.** With an image feature map $x$, we perform Tree Scanning Algorithm (TSA) to construct a 4-connected graph with edge weights measured by dissimilarity between pixels. Then, we obtain an MST with vertices set $\Omega$ through a pruning algorithm and perform the state transition for each vertex in this topology (detailed in Sec. 3.2). Red arrows describe the propagation source of vertex $i$.

parameters $(\bar{\mathbf{A}}, \bar{\mathbf{B}}, \bar{\mathbf{C}}, \bar{\mathbf{D}})$ over the specified sampling time-scale $\Delta \in \mathbb{R}^D$:

$$\bar{\mathbf{A}} = e^{\Delta \mathbf{A}}, \quad \bar{\mathbf{B}} = \left(e^{\Delta \mathbf{A}} - I\right)\mathbf{A}^{-1}\mathbf{B}, \quad \bar{\mathbf{C}} = \mathbf{C}, \quad \bar{\mathbf{D}} = \mathbf{D} \quad (2)$$

In addition, many improved methods [41, 19] use an approximation of $\bar{\mathbf{B}}$ based on the first-order Taylor Series:

$$\bar{\mathbf{B}} = \left(e^{\Delta \mathbf{A}} - I\right)\mathbf{A}^{-1}\mathbf{B} \approx (\Delta \mathbf{A})(\Delta \mathbf{A})^{-1}\Delta \mathbf{B} = \Delta \mathbf{B} \quad (3)$$

**Selective Mechanism .** Previous SSMs store information through finite states and inherent time-invariance, which limits their effectiveness. Therefore, Mamba [19] introduces a dynamic mechanism to selectively filter out input into a sequential state. Specifically, it utilizes Linear Projection to calculate the parameters $\{\mathbf{B}_i\}_{i=1}^L$, $\{\mathbf{C}_i\}_{i=1}^L$ and $\{\Delta_i\}_{i=1}^L$ from the input sequence $\{x_i\}_{i=1}^L$ with $x_i \in \mathbb{R}^{1 \times D}$ directly to improve the context-aware ability. Then the output sequence $\{y_i\}_{i=1}^L$ can be computed with those input-adaptive discretized parameters as follows:

$$h_i = \bar{\mathbf{A}}_i h_{i-1} + \bar{\mathbf{B}}_i x_i, \quad y_i = \mathbf{C}_i h_i + \mathbf{D}x_i \quad (4)$$

## 3.2 Tree State Space Model

Mamba [19] has showcased remarkable performance in modeling the dependencies of consecutive words in a sequence. However, its applicability in long-context tasks, especially visual modeling, still poses certain challenges. For visual tasks, many methods attempt to address this problem by employing fixed scanning strategies, such as multi-directional raster scan [41, 77], local scan [34], and continuous scan [69]. However, these handcrafted scanning methods fail to effectively preserve the 2D structural information of images.

Following the design in Mamba [19], we construct a transform block as a tree state space model, which is presented in Fig. 2. The only difference between our block and Mamba lies in the replacement of the structured state space block with the proposed tree scanning algorithm. In the tree scanning algorithm, we generate a tree topology and then propagate the state of each vertex along the topological path to obtain strong feature representations. In addition, our algorithm can effectively enhance language representations by incorporating such a tree topology during text processing, which overcomes the geometrical constraints of text sequences. In the following, we elaborate on the proposed tree scanning algorithm and its applications for multi-modal tasks.

**Tree Scanning Algorithm.** Given an input feature $X = \{x_i\}_{i=1}^L$ where $L$ is the sequence length (or the number of input pixels), we construct an undirected $m$-connected graph $G = (V, E)$ for the

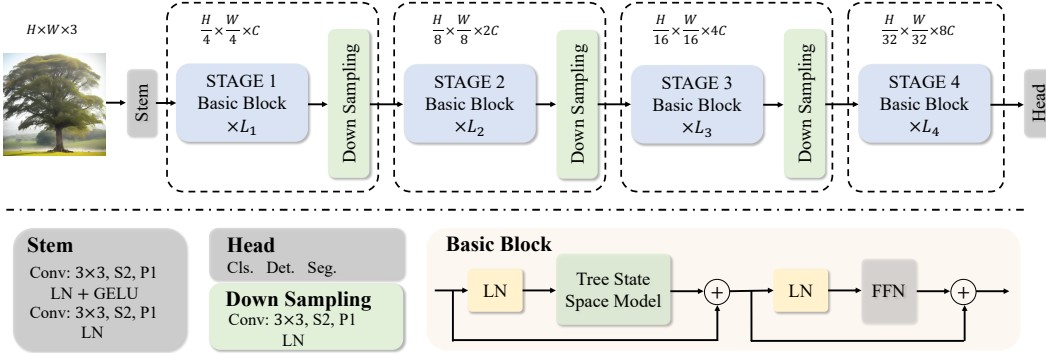

Figure 3: **Overview of MambaTreeV.** LN means LayerNorm and FFN is a feed-forward network in the basic block. S2 and P1 denote stride of 2 and padding size of 1 in convolution, respectively.

feature. $m$ is a hyper-parameter that indicates the number of adjacent tokens. Following [71, 54], we set $m = 4$ for visual tasks, meaning each pixel is connected to its four neighboring pixels. For language tasks, we set $m = 3$ by default, meaning each token is connected to the previous three tokens. In addition, the vertices $V$ represent the pixel (or token) embeddings, and the $E$ indicates the edges of the graph. The edge weight is calculated by the feature dissimilarity between adjacent vertices. Besides, the metric of dissimilarity uses cosine distance by default, and the comparison with other metrics refers to Table 5.

We use the Contractive Boruvka algorithm [2] to prune the edges with significant dissimilarity, which generates a minimum spanning tree (MST) $\mathcal{G}_T$ whose sum of dissimilarity weights is minimum out of all spanning trees. In the propagation process, we iteratively traverse each vertex, treating it as the root, and aggregate the features of the remaining vertices. Intuitively, applying state propagation within such a geometric configuration makes its preferential interactions among vertices with small spatial and feature distances. Following the Mamba, we employ the data-dependent transition matrix for state propagation. For a vertex $k$, we denote the transition matrix with its parent as $\bar{\mathbf{A}}_k$. Furthermore, following the Eq. (4), the state aggregation process for the $i$-th vertex can be formulated as:

$$h_i = \sum_{\forall j \in \Omega} S(E_{ij})\bar{\mathbf{B}}_j x_j, \quad S(E_{ij}) = \prod_{k \in N_{ij}} \bar{\mathbf{A}}_k, \tag{5}$$

where $\Omega$ denotes the index set of all vertices in the tree. $S(E_{ij})$ represents the path weight of hyperedge $E_{ij}$ traced from $j$-th vertex to $i$-th vertex in the tree $\mathcal{G}_T$, and $N_{ij}$ indicates the index set of all vertices on this hyperedge. For visual tasks, we iterate over each vertex, treating it as the root of the spanning tree $\mathcal{G}_T$, and aggregate the states from the other vertices, thereby obtaining the transformed states $\{h_i\}_{i=1}^L$. For textual tasks, because of the causal prediction manner in large language models, we only take the last token as root and aggregate from other tokens. To achieve end-to-end training, we derive the derivative of the output hidden state $h_i$ to the input variables $\bar{\mathbf{A}}_k$, $\bar{\mathbf{B}}_j$ and $x_j$ as follows:

$$\frac{\partial h_i}{\partial x_j} = S\left(E_{ij}\right)\bar{\mathbf{B}}_j, \quad \frac{\partial h_i}{\partial \bar{\mathbf{B}}_j} = S\left(E_{ij}\right)x_j \tag{6}$$

$$\frac{\partial h_i}{\partial \bar{\mathbf{A}}_k} = \sum_{\forall j \in C_k^i} \bar{\mathbf{B}}_j x_j S(E_{kj})S(E_{in}), \tag{7}$$

where $C_k^i$ indicates the children of vertex $k$ in tree $\mathcal{G}_T$ whose root is the vertex $i$, and $n$ denotes the parent of vertex $k$ in Eq. (7). Finally, the output feature $Y$ can be formulated as:

$$Y = \mathbf{C} \odot Norm(H) + \mathbf{D} \odot X, \tag{8}$$

where $Y$, $H$ and $X$ indicate the stack of $\{y_i\}_{i=1}^L$, $\{h_i\}_{i=1}^L$ and $\{x_i\}_{i=1}^L$ respectively. $\odot$ denotes the element-wise multiplication.

**Efficient Implementation for Multi-Modality.** For visual tasks, the tree scanning algorithm requires two levels of traversal across the entire pixel set, resulting in an unacceptable quadratic complexity relative to the number of pixels $\mathcal{O}(L^2)$. To alleviate this issue, we utilize a dynamic

**Algorithm 1** Vision Tree Scanning

---

**Input:** Input feature $\{x_i\}_{i=1}^L$; Input matrix $\{\bar{\mathbf{B}}_i\}_{i=1}^L$; State matrix $\{\bar{\mathbf{A}}_i\}_{i=1}^L$; Gradient of loss to hidden states $\{\frac{\partial Loss}{\partial h_i}\}_{i=1}^L$; Minimum Spanning Tree $\mathcal{G}_T$.

**Traverse Path:** $Root, \dots, Leaf \leftarrow BFS(\mathcal{G}_T)$      ▷ *Breadth-first topological order of $\mathcal{G}_T$*

**Forward:**

     Initialization: $\{\xi_i\}_{i=1}^L \leftarrow \{x_i\}_{i=1}^L$

2: **for** $i \leftarrow Leaf$ to $Root$ **do**

     $\xi_i = \bar{\mathbf{B}}_i x_i + \sum_{\forall j \in \{t | \text{Par}(t) = i\}} \xi_j \bar{\mathbf{A}}_j$

4: **end for**

     **for** $i \leftarrow Root$ to $Leaf$ **do**

6:      **if** $i$ is $Root$ **then**

         $h_i = \xi_i$

8:      **else**

         $h_i = \bar{\mathbf{A}}_i(h_{\text{Par}(i)} - \bar{\mathbf{A}}_i \xi_i) + \xi_i = (1 - \bar{\mathbf{A}}_i^2)\xi_i + \bar{\mathbf{A}}_i h_{\text{Par}(i)}$

10:      **end if**

     **end for**

**Backward:**

12: Initialization: $\{\eta_i\}_{i=1}^L \leftarrow \{\frac{\partial Loss}{\partial h_i}\}_{i=1}^L$

     **for** $i \leftarrow Leaf$ to $Root$ **do**

14:      $\eta_i = \bar{\mathbf{B}}_i \frac{\partial Loss}{\partial h_i} + \sum_{\forall j \in \{t | \text{Par}(t) = i\}} \eta_j \bar{\mathbf{A}}_j$

     **end for**

16: **for** $i \leftarrow Root$ to $Leaf$ **do**

     **if** $i$ is $Root$ **then**

18:      $\frac{\partial Loss}{\partial x_i} = \eta_i \bar{\mathbf{B}}_i$ ,     $\frac{\partial Loss}{\partial \bar{\mathbf{B}}_i} = \eta_i x_i$ ,     $\frac{\partial Loss}{\partial \bar{\mathbf{A}}_i} = 0$

     **else**

20:      $\frac{\partial Loss}{\partial x_i} = (1 - \bar{\mathbf{A}}_i^2)\eta_i \bar{\mathbf{B}}_i + \bar{\mathbf{A}}_i \frac{\partial Loss}{\partial x_{\text{Par}(i)}} \bar{\mathbf{B}}_i$ ,     $\frac{\partial Loss}{\partial \bar{\mathbf{B}}_i} = (1 - \bar{\mathbf{A}}_i^2)\eta_i x_i + \bar{\mathbf{A}}_i \frac{\partial Loss}{\partial \bar{\mathbf{B}}_{\text{Par}(i)}} x_i$

     $\frac{\partial Loss}{\partial \bar{\mathbf{A}}_i} = \eta_i * (h_i - \bar{\mathbf{A}}_i \xi_i) + \xi_i * (\frac{\partial Loss}{\partial x_i} - \bar{\mathbf{A}}_i \eta_i) = \eta_i h_i + \xi_i \frac{\partial Loss}{\partial x_i} - 2\eta_i \xi_i \bar{\mathbf{A}}_i$

22:      **end if**

     **end for**

**Output:** Hidden states $\{h_i\}_{i=1}^L$; Grad. of loss to input feature $\{\frac{\partial Loss}{\partial x_i}\}_{i=1}^L$; Grad. of loss to input matrix $\{\frac{\partial Loss}{\partial \bar{\mathbf{B}}_i}\}_{i=1}^L$; Grad. of loss to state matrix $\{\frac{\partial Loss}{\partial \bar{\mathbf{A}}_i}\}_{i=1}^L$.

---

programming procedure to accelerate the inference and training processes as elaborated in Algorithm 1, which results in linear complexity $\mathcal{O}(L)$. For textual tasks, we perform a unidirectional aggregation approach (shown in Algorithm 2 of Appendix B) in adherence to the causal nature of language. Moreover, we provide the back-propagation process for both Vision Tree Scanning and Language Tree Scanning processes, whose detailed proofs refer to Appendix C.

### 3.3 Application for Vision and Language

**MambaTreeV**   Given an image with a shape of $H \times W \times 3$, our goal is to obtain high-quality visual features for downstream tasks. To this end, we propose an effective vision architecture MambaTreeV which consists of a stem module, several basic blocks and downsampling layers to generate hierarchical representations illustrated in Fig. 3. Overall, our MambaTreeV comprises four stages similar to previous general vision backbones [44, 43, 62, 41]. We integrate the stem module before the first stage to decrease the resolution of the input image signal by a factor of $4$, resulting in a feature map with a shape of $\frac{H}{4} \times \frac{W}{4} \times C$. It includes two convolutions, two Layer Normalization (LN) layers and one GELU activation function. The kernel size for both convolutions is 3 with a stride of $2$ and padding of $1$. Similarly, a downsampling layer consists of a $3 \times 3$ convolution with a stride of $2$ and padding of $1$ and an LN layer. Positioned between two stages, it serves to downsample the input feature map by a factor of $2$. Motivated by [62, 41], we devise a residual block with skip connections to integrate our fundamental Tree State Space Model in Sec. 3.2. In detail, we first normalize the input features with LN layer. Spatial priors and long-range dependencies are then obtained through our tree scanning algorithm with residual connections established alongside the input features. Finally, a feedforward neural network is utilized to project the normalized features to output signals as shown

| Method | Type | #Param. | #FLOPs | Top-1 Acc. | Method | Type | #Param. | #FLOPs | Top-1 Acc. |
|---|---|---|---|---|---|---|---|---|---|
| Deit-S [59] | T | 22M | 4.6G | 79.9 | ConvNeXt-S [44] | C | 50M | 8.7G | 83.1 |
| Swin-T [43] | T | 28M | 4.6G | 81.3 | SLaK-S [40] | C | 55M | 9.8G | 83.8 |
| CoAtNet-0 [11] | T | 25M | 4.0G | 81.6 | UniRepLKNet-S [13] | C | 56M | 9.1G | 83.9 |
| SG-Former-S [50] | T | 23M | 4.8G | 83.2 | InternImage-S [62] | C | 50M | 8.0G | 84.2 |
| ConvNeXt-T [44] | C | 29M | 4.5G | 82.1 | HyenaViT-B [17] | S | 88M | - | 78.5 |
| SLaK-T [40] | C | 30M | 5.0G | 82.5 | S4ND-ViT-B [48] | S | 89M | - | 80.4 |
| UniRepLKNet-T [13] | C | 31M | 4.9G | 83.2 | PlainMamba-L3 [69] | S | 50M | 14.4G | 82.3 |
| InternImage-T [62] | C | 30M | 5.0G | 83.5 | VMamba-S [41] | S | 50M | 8.7G | 83.6 |
| ViM-S [77] | S | 26M | 5.1G | 80.5 | LocalVMamba-S [34] | S | 50M | 11.4G | 83.7 |
| LocalViM-S [34] | S | 28M | 4.8G | 81.2 | MambaTreeV-S (Ours) | S | 51M | 8.5G | **84.2** |
| PlainMamba-L2 [69] | S | 25M | 8.1G | 81.6 | Deit-B [59] | T | 86M | 55.4G | 83.1 |
| Mamba-2D-S [37] | S | 24M | - | 81.7 | Swin-B [43] | T | 88M | 15.4G | 83.5 |
| S4ND-ConvNeXt-T [48] | S | 30M | - | 82.2 | CoAtNet-2 [11] | T | 75M | 16.0G | 84.1 |
| VMamba-T [41] | S | 31M | 4.9G | 82.5 | ConvNeXt-B [44] | C | 89M | 15.4G | 83.8 |
| LocalVMamba-T [34] | S | 26M | 5.7G | 82.7 | SLaK-B [40] | C | 95M | 17.0G | 84.0 |
| MambaTreeV-T (Ours) | S | 30M | 4.8G | **83.4** | Mamba-2D-B [37] | S | 92M | - | 83.0 |
| Swin-S [43] | T | 50M | 8.7G | 83.0 | VMamba-B [41] | S | 89M | 15.4G | 83.9 |
| CoAtNet-1 [11] | T | 42M | 8.0G | 83.3 | MambaTreeV-B (Ours) | S | 91M | 15.1G | **84.8** |

Table 1: **Image classification performance on the ImageNet-1K validation set.** T, C and S indicate the model type of Transformer, CNN and SSM, respectively. All models take a scale of $224^2$ as input.

in Fig. 3. Based on the above origin components, we develop our MambaTreeV in three scales, *i.e.*, MambaTreeV-Tiny, MambaTreeV-Small and MambaTreeV-Base.

**MambaTreeL** Recurrent neural networks rely on fixed memory to preserve past information, which poses limitations when handling long contexts where relevant words are distant from the current moment. While Mamba [19] employs a selection mechanism to enhance context awareness, its fixed memory size cannot expand over time, resulting in restricted state space. Therefore, the ability to extrapolate decreases during scrolling as the prompt extends. To mitigate this issue, we propose an effective fine-tuning paradigm. Specifically, the tree-based topology branch is built upon one-way scrolling with a scaling factor, enabling state transitions within such a structure. This arrangement facilitates the preferential interaction of semantically related tokens. It is noteworthy that this paradigm does not introduce any additional training parameters. Instead, it utilizes pretrained state transformation parameters to conduct semantic aggregation by incorporating topological structures. Experimental results demonstrate the effectiveness of our approach.

## 4 Experiments

We conduct extensive experiments to evaluate the effectiveness of MambaTreeV and compare it with advanced CNN-based, Transformer-based, and SSM-based models covering various downstream tasks, including image classification, object detection and semantic segmentation. Furthermore, we validate the capability of MambaTreeL in the field of natural language understanding.

### 4.1 Image Classification

**Settings.** We assess the classification performance of MambaTreeV on the ImageNet-1k dataset [12]. Following previous practices [43, 44, 62, 41], all MambaTreeV models are trained for 300 epochs from scratch using AdamW optimizer with a warm-up strategy of 20 epochs. During training, we utilize a Cosine Scheduler with an initial learning rate of $1 \times 10^{-3}$ and weight decay of 0.05. In addition, the exponential moving average (EMA) is also applied.

**Results.** The comparison results summarized in Table 1 show MambaTreeV leading all SSM-based models and competitive with advanced CNNs and Transformers across tiny, small, and base scales. Specifically, MambaTreeV-T achieves 83.4% Top-1 Acc. boosting ViM-S by 2.9%, LocalVim-S by 2.2%, PlainMamba-L2 by 1.8% and VMamba-T by 0.9% with similar FLOPs. Additionally, it surpasses ConvNeXt-T by 1.3% and Swin-T by 2.2%, demonstrating the effectiveness of our method.

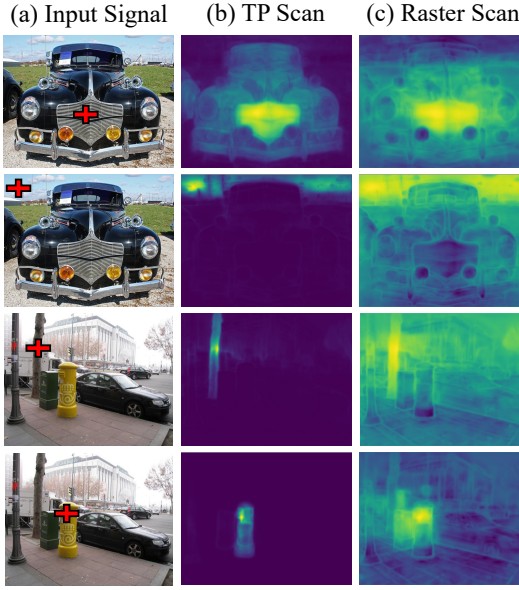

| (a) Input Signal | (b) TP Scan | (c) Raster Scan |
|---|---|---|

Figure 4: **Visualization of affinity maps in the specific position.** The Location is marked by the red cross in each input (a). TP is our tree topology scanning algorithm (b), which captures more detailed structural information and has a larger receptive field compared to raster scanning (c).

| Method | Type | #FLOPs | mIoU SS | mIoU MS |
|---|---|---|---|---|
| Swin-T [43] | T | 945G | 44.5 | 45.8 |
| ConvNeXt-T [44] | C | 939G | 46.0 | 46.7 |
| SLaK-T [40] | C | 936G | 47.6 | - |
| InternImage-T [62] | C | 944G | 47.9 | 48.1 |
| UniRepLKNet-T [13] | C | 946G | 48.6 | 49.1 |
| ViM-S [77] | S | - | 44.9 | - |
| LocalViM-S [34] | S | 297G | 46.4 | 47.5 |
| PlainMamba-L2 [69] | S | 285G | 46.8 | - |
| VMamba-T [41] | S | 964G | 47.3 | 48.3 |
| LocalVMamba-T [41] | S | 970G | 47.9 | 49.1 |
| MambaTreeV-T (Ours) | S | 941G | **48.5** | **49.4** |
| Swin-S [43] | T | 1038G | 47.6 | 49.5 |
| ConvNeXt-S [44] | C | 1027G | 48.7 | 49.6 |
| SLaK-S [40] | C | 1028G | 49.4 | - |
| InternImage-S [62] | C | 1017G | 50.1 | 50.9 |
| UniRepLKNet-S [13] | C | 1036G | 50.5 | 51.0 |
| PlainMamba-L3 [69] | S | 419G | 49.1 | - |
| VMamba-S [41] | S | 1081G | 49.5 | 50.5 |
| LocalVMamba-S [34] | S | 1095G | 50.0 | 51.0 |
| MambaTreeV-S (Ours) | S | 1019G | **50.7** | **51.7** |

Table 2: **Semantic segmentation performance on ADE20K val set.** The crop size is all set to $512^2$. SS and MS denote single-scale and multi-scale testing, respectively.

## 4.2 Object Detection

**Settings.** We verify the detection performance of MambaTreeV on the MSCOCO 2017 dataset [39] with MMDetection library [3]. We follow previous works [41, 62, 43, 34, 53, 55, 74, 70, 6] to validate object detection and instance segmentation tasks with Mask-RCNN [29]. Specifically, We adopt the AdamW optimizer with a learning rate of $1 \times 10^{-4}$ and batch size of 16 to optimize the model built upon our pre-trained classification backbones on ImageNet-1K. The training schedules include $1\times$ (12 epochs) and $3\times$ (36 epochs) with multi-scale data augmentation.

**Results.** As depicted in Table 8 (in Appendix A.), our method outperforms existing methods on most evaluation metrics, especially for instance segmentation. Under $1\times$ schedule, MambaTreeV-T achieves $47.0$ in box mAP ($AP^b$), which is $1.1$ points higher than ViM-S and $0.5$ points higher than VMamba-T. It is worth noting that MambaTreeV-T outperforms ViM-S by $1.7$ points with $1\times$ schedule and LocalVMamba-T by $0.4$ points with $3\times$ schedule in mask mAP ($AP^m$). Moreover, the best $AP^b$ $50.1$ and $AP^m$ $44.6$ are obtained by MambaTreeV-S in $3\times$ schedule with multi-scale training.

## 4.3 Semantic Segmentation

**Settings.** To evaluate the semantic segmentation performance of our MambaTreeV series, we train our models with UperNet [65] initialized by pre-trained classification weights on ADE20K[75] for 160k iterations, following common practices without additional augmentations for fair comparison.

**Results.** Our method performs exceptionally well on segmentation tasks shown in Table 2. MambaTreeV-T yields a clear improvement of $+3.6$ in single-scale mIoU compared to ViM-S and $+1.9$ in multi-scale mIoU compared to LocalViM-S. Furthermore, MambaTreeV-S boosts InternImage-S by $0.6$ and $0.8$ in single-scale and multi-scale respectively. We consider the preservation of intricate structural details through tree topology scanning to be particularly advantageous for segmentation tasks that require pixel-level perception.

| Method | PIQA ↑ | Arc-E ↑ | SST ↑ | WG ↑ | L-ppl ↓ | Race ↑ | BQA ↑ | Average Acc. ↑ |
|---|---|---|---|---|---|---|---|---|
| Mamba [19] | 64.5 | 48.0 | 65.6 | 51.8 | 16.1 | 27.4 | 16.8 | 45.7 |
| + LoRA [33] | 64.7 | 48.3 | 65.1 | **52.2** | 17.7 | 28.6 | 17.8 | 46.1 |
| + MambaTreeL (Ours) | **65.0** | **49.8** | **69.5** | 51.1 | **15.9** | **28.9** | **19.2** | **47.2** |

Table 3: **Evaluation on language model benchmarks.** Arc-E, WG, L-ppl and BQA indicate Arc-easy [8], WinoGrande, LAMBADA [49] and Openbookqa [47] benchmark, respectively.

| Scanning Strategy | Acc |
|---|---|
| Raster Scan | 82.6 |
| Cross Scan | 83.1 |
| Tree Topology Scan | **83.4** |

| Distance Metric | Acc. |
|---|---|
| *Manhattan* | 82.9 |
| *Euclidean* | 83.2 |
| *Cosine* | **83.4** |

| Root Setting | Acc. |
|---|---|
| First vertex | 82.9 |
| Last vertex | 83.0 |
| All vertices | **83.4** |

Table 4: **Effectiveness of our algorithm.**

Table 5: **Impact of different distance Metrics.**

Table 6: **Superiority of traversing all vertices.**

## 4.4 Language Understanding

We regard Mamba [19] with 130M parameters as the base model. To verify the effectiveness of our MambaTreeL in nature language understanding, we first fine-tune pre-trained Mamba via LoRA [33] and MambaTreeL under the same setting with the Alpaca data [58], which contains 52000 instruction tuning data for supervised fine-tuning. Then we utilize popular language benchmarks provided in the open-sourced lm-evaluation-harness project [18] for evaluation, including PIQA [1], AI2-ARC [8], SST [60], WinoGrande, LAMBADA [49], Race [36] and Openbookqa [47]. The results in Table 3 demonstrate that our MambaTreeL provides a benefit of $+1.1\%$ in average Acc. compared to LoRA. Since the short prompt length of WinoGrande dataset, the performance degrades with a marginal gap.

## 4.5 Ablation Study & Qualitative Results

In this section, we conduct analysis experiments on ImageNet-1K dataset and present some visual results to illustrate the effectiveness of our algorithm.

**Scanning Strategy.** We conduct a head-to-head comparison of different scanning strategies, as shown in Table 4. The tree topology scanning outperforms previous strategies by $0.8\%$ and $0.3\%$, highlighting the superiority of our algorithm in vision recognition.

**Distance Metric.** Before generating a minimum spanning tree from a connected graph, it is important to measure the edge weights between vertices. Therefore, we validate several distance metrics as illustrated in Table 5. The results indicate that $Cosine$ distance most effectively represents the relationship between vertices, performing $0.5\%$ better than $Manhattan$ and $0.2\%$ better than $Euclidean$.

**Root Setting.** We traverse all vertices, treating each as a root, and perform state transitions along the topological path from the other vertices toward the root. This traversal ensures that each vertex captures long-range dependencies. To verify the effectiveness of this operation, we consider only the first and last vertices as the root in Table 6. The results show reductions of $0.5\%$ and $0.4\%$, respectively.

**Inference speed comparison.** As shown in Table 7, we report the inference throughputs of our method on an Nvidia V100 GPU. MambaTreeV-T* refers to each stage sharing the same tree topology structure, which enhances efficiency without compromising accuracy. To achieve better practical inference speed, we also introduce a cuda implementation optimized for GPUs. Compared with other counterparts, our approach exhibits superior effectiveness and faster inference speed.

**Qualitative Results.** To better illustrate the superiority of our scanning strategy, we visualize the affinity maps of different positions marked by the red cross in each input image. For example, we

| Method (224x224) | Throughput (img/s) | GPU Memory | FLOPs | #Params. | Acc. (Top1.) |
|---|---|---|---|---|---|
| PlainMamba-L2 [69] | 363 | 4204M | 8.1G | 25M | 81.6 |
| VMamba-T [41] | 374 | 8646M | 4.9G | 31M | 82.5 |
| LocalVMamba-T [34] | 311 | 11298M | 5.7G | 26M | 82.7 |
| MambaTreeV-T(one root) | 283 | 6012M | 4.8G | 30M | 83.0 |
| MambaTreeV-T | 281 | 6471M | 4.8G | 30M | 83.4 |
| MambaTreeV-T* | **392** | **4800M** | **4.8G** | 30M | **83.4** |

Table 7: **Runtime comparison on an Nvidia V100 GPU during inference.**

set the anchor point in the upper left corner of the sky as shown in the second row of in Fig. 4(a). Our method can easily identify white houses, flagpoles, and the sky, which raster scanning fails to achieve. This demonstrates the capability of our algorithm to preserve detailed structural information. More comparisons can be seen in Fig. 6 (in Appendix D.)

## 5    Conclusion & Limitations

In this paper, we propose a tree state space model to perform feature propagation on an input-aware topology. Besides, we introduce a linear complexity dynamic programming algorithm to enhance long-range interactions without increasing computational cost. With the proposed techniques, we establish the general multi-modal networks to break the original sequence constraints and achieve stronger representation capabilities. Extensive experiments demonstrate the effectiveness of our method in both visual and language tasks. The limitation of our method is that the tree structure is not a common paradigm, and it needs to be specifically optimized according to the hardware device.

## Acknowledgments and Disclosure of Funding

This work was supported by the STI 2030-Major Projects under Grant 2021ZD0201404.

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

# Appendix

## A    Detailed Training Settings and Results

### A.1    Image Classification.

We follow the previous works [62, 41, 43] to conduct the experiments. The models are trained with thirty-two 32GB V100 GPUs by default. We set betas and momentum of the AdamW [45, 76, 66] optimizer with $(0.9, 0.999)$ and $0.9$, respectively. During training, we utilize a Cosine Scheduler with an initial learning rate of $1 \times 10^{-3}$ and weight decay of $0.05$. We adopt the common training data augmentation strategies following [34, 62], including AutoAugment [9] with $rand$-$m9$-$mstd0.5$-$inc1$. A MixUp strategy with a ratio of $0.8$ is also adopted in each batch. Horizontal flip and Random resized crop strategy are both used in the process of training.

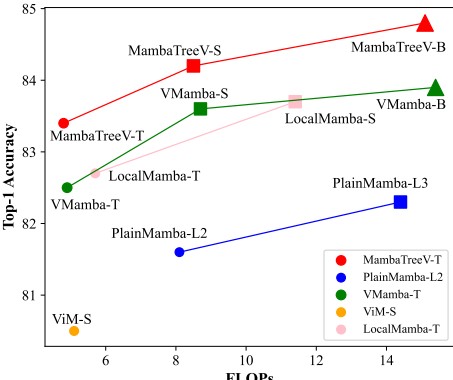

Figure 5: **Classification performance comparison among SSM-based vision foundation models.**

**Performance Comparison.**    We compare various SSM-based visual foundation models as shown in Fig. 5, with different colors representing different models and different shapes indicating different model scales. The size of each shape indicates the number of model parameters. The horizontal axis denotes FLOPs and the vertical axis represents the Top-1 accuracy of the corresponding method on ImageNet-1K val dataset. Fig. 5 demonstrates that MambaTreeV is the best choice in terms of efficiency and effectiveness.

### A.2    Object Detection.

For a fair comparison, we conduct the evaluation following common practice [62, 41, 43]. The models are trained with eight 32GB V100 GPUs by default. The input image is resized so that the shorter side is 800 pixels, while the longer side does not exceed 1333 pixels during the $1\times$ schedule. The number of warmup steps is set to $500$ in the $1\times$ schedule. For $3\times$ schedule, the shorter side is resized to 480-800 pixels and the longer side does not exceed 1333 pixels. The number of warmup steps is set to $1000$ in $3\times$ schedule. Results shown in Table 8 demonstrate the effectiveness of MambaTreeV in object detection and instance segmentation on COCO val2017.

### A.3    Semantic Segmentation.

We optimize our MambaTreeV-T/S using AdamW optimizer with an initial learning rate of $6 \times 10^{-5}$ which is decayed by a rate of $1.0$ with the polynomial decay schedule following [62, 27, 57, 16, 28]. The number of warmup iters is set to 1600 with an initial learning rate of $1 \times 10^{-6}$ [41, 34, 46, 67]. The default input resolution is $512 \times 512$ as well as FLOPs are calculated with an input size of $512 \times 2048$. The models are trained with eight 32GB V100 GPUs by default.

| Method | #FLOPs. | Mask R-CNN 1× Schedule | | | | | | Mask R-CNN 3× MS Schedule | | | | | |
|---|---|---|---|---|---|---|---|---|---|---|---|---|---|
| | | $AP^b$ | $AP^b_{50}$ | $AP^b_{75}$ | $AP^m$ | $AP^m_{50}$ | $AP^m_{75}$ | $AP^b$ | $AP^b_{50}$ | $AP^b_{75}$ | $AP^m$ | $AP^m_{50}$ | $AP^m_{75}$ |
| Swin-T [43] | 267G | 42.7 | 65.2 | 46.8 | 39.3 | 62.2 | 42.2 | 46.0 | 68.1 | 50.3 | 41.6 | 65.1 | 44.9 |
| ConvNeXt-T [44] | 262G | 44.2 | 66.6 | 48.3 | 40.1 | 63.3 | 42.8 | 46.2 | 67.9 | 50.8 | 41.7 | 65.0 | 44.9 |
| CSWin-T [14] | 279G | 46.7 | 68.6 | 51.3 | 42.2 | 65.6 | 45.4 | 49.0 | 70.7 | 53.7 | 43.6 | 67.9 | 46.6 |
| ViM-S [77] | 218G | 44.9 | 67.1 | 49.3 | 41.0 | 64.2 | 44.1 | - | - | - | - | - | - |
| VMamba-T [41] | 286G | 46.5 | 68.5 | 50.7 | 42.1 | 65.5 | 45.3 | 48.5 | 69.9 | 52.9 | 43.2 | 66.8 | 46.3 |
| L-Vmamba-T [34] | 291G | 46.7 | 68.7 | 50.8 | 42.2 | 65.7 | 45.5 | 48.7 | 70.1 | 53.0 | 43.4 | 67.0 | 46.4 |
| MambaTreeV-T (Ours) | 265G | **47.0** | **69.4** | **51.5** | **42.7** | **66.4** | **46.0** | **49.0** | **70.8** | **54.0** | **43.8** | **67.6** | **47.1** |
| Vit-Adapter-S [4] | 403G | 44.7 | 65.8 | 48.3 | 39.9 | 62.5 | 42.8 | 48.2 | 69.7 | 52.5 | 42.8 | 66.4 | 45.9 |
| Swin-S [43] | 354G | 44.8 | 66.6 | 48.9 | 40.9 | 63.4 | 44.2 | 48.2 | 69.8 | 52.8 | 43.2 | 67.0 | 46.1 |
| ConvNeXt-T [44] | 348G | 45.4 | 67.9 | 50.0 | 41.8 | 65.2 | 45.1 | 47.9 | 70.0 | 52.7 | 42.9 | 66.9 | 46.2 |
| InternImage-S [62] | 340G | 47.8 | 69.8 | 52.8 | 43.3 | 67.1 | 46.7 | 49.7 | 71.1 | 54.5 | 44.5 | 68.5 | 47.8 |
| VMamba-S [41] | 400G | 48.2 | 69.7 | 52.5 | 43.0 | 66.6 | 46.4 | 49.7 | 70.4 | 54.2 | 44.0 | 67.6 | 47.3 |
| L-Vmamba-S [34] | 414G | 48.4 | 69.9 | 52.7 | 43.2 | 66.7 | 46.5 | 49.9 | 70.5 | 54.4 | 44.1 | 67.8 | 47.4 |
| MambaTreeV-S (Ours) | 341G | **48.6** | **70.3** | **53.5** | **43.6** | **67.5** | **47.1** | **50.1** | **71.2** | **54.9** | **44.6** | **68.7** | **47.8** |

Table 8: **Object detection and instance segmentation performance on COCO val2017.** $AP^b$ and $AP^m$ indicate the mAP of detection and segmentation, respectively. MS indicates the multi-scale training strategy.

# B  Language Tree Topology Scanning Operator

---

**Algorithm 2** Language Tree Scanning

---

**Input:** Input feature $\{x_i\}_{i=1}^L$; Input matrix $\{\bar{\mathbf{B}}_i\}_{i=1}^L$; State matrix $\{\bar{\mathbf{A}}_i\}_{i=1}^L$; Gradient of loss to hidden states $\{\frac{\partial Loss}{\partial h_i}\}_{i=1}^L$; Minimum Spanning Tree $\mathcal{G}_T$.

**Traverse Path:** $Root, \ldots, Leaf \leftarrow BFS(\mathcal{G}_T)$  ▷ *Breadth-first topological order of $\mathcal{G}_T$*

**Forward:**

    Initialization: $\{\xi_i\}_{i=1}^L \leftarrow \{x_i\}_{i=1}^L$

2: **for** $i \leftarrow Leaf$ to $Root$ **do**

    $\xi_i = \bar{\mathbf{B}}_i x_i + \sum_{\forall j \in \{t|\text{Par}(t)=i\}} \xi_j \bar{\mathbf{A}}_j$

4: **end for**

**Backward:**

    **for** $i \leftarrow Root$ to $Leaf$ **do**

6:     **if** $i$ is $Root$ **then**

        $\frac{\partial Loss}{\partial x_i} = \eta_i \bar{\mathbf{B}}_i$ ,     $\frac{\partial Loss}{\partial \bar{\mathbf{B}}_i} = \eta_i x_i$,     $\frac{\partial Loss}{\partial \bar{\mathbf{A}}_i} = 0$

8:     **else**

        $\frac{\partial Loss}{\partial x_i} = \frac{\partial Loss}{\partial h_i}\bar{\mathbf{B}}_i + \bar{\mathbf{A}}_i \frac{\partial Loss}{\partial x_{\text{Par}(i)}}\bar{\mathbf{B}}_i$ ,     $\frac{\partial Loss}{\partial \bar{\mathbf{B}}_i} = \frac{\partial Loss}{\partial h_i}x_i + \bar{\mathbf{A}}_i \frac{\partial Loss}{\partial \bar{\mathbf{B}}_{\text{Par}(i)}}x_i$

10:         $\frac{\partial Loss}{\partial \bar{\mathbf{A}}_i} = \frac{\partial Loss}{\partial x'_{Par(i)}}h_i$

    **end if**

12: **end for**

**Output:** Hidden states $\{h_i\}_{i=1}^L$; Grad. of loss to input feature $\{\frac{\partial Loss}{\partial x_i}\}_{i=1}^L$; Grad. of loss to input matrix $\{\frac{\partial Loss}{\partial \bar{\mathbf{B}}_i}\}_{i=1}^L$; Grad. of loss to state matrix $\{\frac{\partial Loss}{\partial \bar{\mathbf{A}}_i}\}_{i=1}^L$.

---

# C  Algorithm Proof

In this section, we present detailed proofs for our tree scanning algorithm. The definitions of symbols are consistent with those in the main paper.

### C.1   Proof for Algorithm 1.

We randomly take a vertex in the MST $\mathcal{G}_T$ as the $root$. According to the definition of the tree scanning algorithm introduced in Sec. 3.2, we can derive $h_{root}$ as follows:

$$h_{root} = \sum_{\forall j \in C_{root}} S(E_{root,j})\bar{\mathbf{B}}_j x_j, \quad S(E_{root,j}) = \prod_{k \in N_{root,j}} \bar{\mathbf{A}}_k, \tag{9}$$

which shows a process of aggregation from all leaf vertices to the $root$. Therefore, each vertex is only related to its child in this period. Taking vertex $m$ as an example, the $\text{Aggr}_m$ can be derived as:

$$\text{Aggr}_m(x) = \bar{\mathbf{B}}_m x_m + \sum_{\forall k \in \{t|\text{Par}(t)=i\}} \text{Aggr}_k(x)\bar{\mathbf{A}}_k. \tag{10}$$

We assume that one of the child of $m$ is $n$ and $h_n$ can be derived as following:

$$h_n = \text{Aggr}_n(x) + \bar{\mathbf{A}}_n \widetilde{\text{Aggr}}_m(x), \tag{11}$$

where $\widetilde{\text{Aggr}}_m(x)$ indicates the aggregation value from the vertices $\in \Omega \setminus C_m^{root}$ to vertex $m$. Therefore, we can obtain the propagation relationship between the hidden state of parent $m$ and child $n$:

$$
\begin{aligned}
h_n &= \text{Aggr}_n(x) + \bar{\mathbf{A}}_n \widetilde{\text{Aggr}}_m(x) \\
&= \text{Aggr}_n(x) + \bar{\mathbf{A}}_n(h_m - \bar{\mathbf{A}}_n \text{Aggr}_n(x)) \\
&= \bar{\mathbf{A}}_n h_m + (1 - \bar{\mathbf{A}}_n^2)\text{Aggr}_n(x)
\end{aligned}
\tag{12}
$$

Through the above derivation, we can calculate $\{h_i\}_{i=1}^L$ with only two traversals (*i.e.*, the aggregation from $leaf$ to $root$ and the propagation from $root$ to $leaf$) in the forward process as shown in Algorithm 1, thereby reducing the computational complexity from $\mathcal{O}(L^2)$ to $\mathcal{O}(L)$.

Next, we analyze the backpropagation process in Algorithm 1. According to the chain rule, we can easily calculate the derivative of $loss$ with respect to $x_i$:

$$
\begin{aligned}
\frac{\partial loss}{\partial x_i} &= \sum_{j \in \Omega} \frac{\partial\, loss}{\partial h_j} \frac{\partial h_j}{\partial x_i} \\
&= \bar{\mathbf{B}}_i \sum_{j \in \Omega} S\left(E_{ji}\right) \frac{\partial loss}{\partial h_j}
\end{aligned}
\tag{13}
$$

Similarly, the derivative of $loss$ with respect to $\bar{\mathbf{B}}_i$ is:

$$
\begin{aligned}
\frac{\partial loss}{\partial \bar{\mathbf{B}}_i} &= \sum_{j \in \Omega} \frac{\partial\, loss}{\partial h_j} \frac{\partial h_j}{\partial \bar{\mathbf{B}}_i} \\
&= x_i \sum_{j \in \Omega} S\left(E_{ji}\right) \frac{\partial loss}{\partial h_j}
\end{aligned}
\tag{14}
$$

The above formulas are equivalent to replacing the input $x$ with $\frac{\partial loss}{\partial h}$ during the forward process.

Subsequently, we assume that the vertex $k$ is the child of vertex $l$ and define $C_l^k$ indicates the children of vertex $l$ with the root of vertex $k$. $\frac{\partial loss}{\partial \bar{\mathbf{A}}_k}$ is formulated as follows:

$$
\begin{aligned}
\frac{\partial loss}{\partial \bar{\mathbf{A}}_k} &= \sum_{j \in \Omega} \frac{\partial loss}{\partial h_j} \frac{\partial h_j}{\partial \bar{\mathbf{A}}_k} \\
&= \sum_{j \in \Omega} \frac{\partial loss}{\partial h_j} \sum_{p \in \Omega} \frac{\partial S(E_{jp}) \bar{\mathbf{B}}_p x'_p}{\partial \bar{\mathbf{A}}_k} \\
&= \sum_{j \in C_l^k} \frac{\partial loss}{\partial h_j} \sum_{p \in C_k^l} S(E_{kp}) S(E_{jl}) \bar{\mathbf{B}}_p x'_p + \sum_{j \in C_k^l} \frac{\partial loss}{\partial h_j} \sum_{p \in C_l^k} S(E_{kj}) S(E_{pl}) \bar{\mathbf{B}}_p x'_p \\
&= \sum_{j \in C_l^k} S(E_{jl}) \frac{\partial loss}{\partial h_j} \sum_{p \in C_k^l} S(E_{kp}) \bar{\mathbf{B}}_p x'_p + \sum_{j \in C_k^l} S(E_{kj}) \frac{\partial loss}{\partial h_j} \sum_{p \in C_l^k} S(E_{pl}) \bar{\mathbf{B}}_p x'_p \\
&= \left( \frac{\partial Loss}{\partial x_k} - \bar{\mathbf{A}}_k \mathrm{Aggr}_k(\frac{\partial loss}{\partial h}) \right) * \mathrm{Aggr}_k(x) + \mathrm{Aggr}_k(\frac{\partial loss}{\partial h}) * (h_k - \bar{\mathbf{A}}_k \mathrm{Aggr}_k(x)) \\
&= \frac{\partial Loss}{\partial x_k} \mathrm{Aggr}_k(x) + \mathrm{Aggr}_k(\frac{\partial loss}{\partial h}) h_k - 2 \bar{\mathbf{A}}_k \mathrm{Aggr}_k(\frac{\partial loss}{\partial h}) \mathrm{Aggr}_k(x) \\
&\triangleq \frac{\partial Loss}{\partial x_k} \xi_k + \eta_k h_k - 2 \bar{\mathbf{A}}_k \eta_k \xi_k \quad \textit{(definition in Algorithm 1)}
\end{aligned}
\tag{15}
$$

So far we have completed the proof of forward and back-propagation of Algorithm 1.

## C.2 Proof for Algorithm 2.

We only take the last token as root and replace the transition source from $\Omega$ to $C_i$ in sequence modeling tasks like nature language understanding to ensure causality. Therefore, only one traversal (from $leaf$ to $root$) is required for the forward process, and another traversal (from $root$ to $leaf$) is needed for the backpropagation process. The proof is similar to the Algorithm 1.

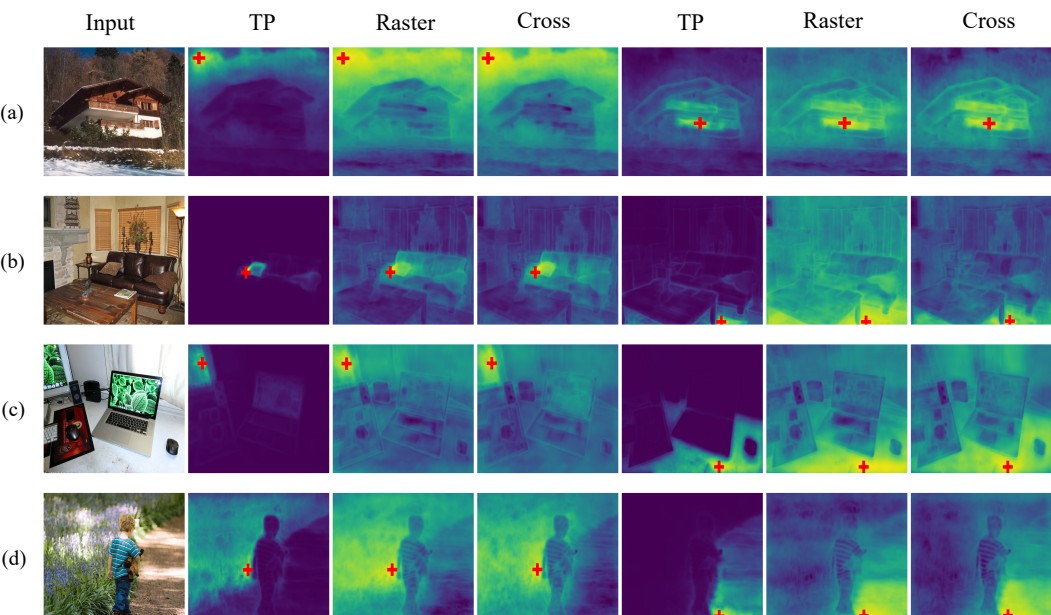

Figure 6: **Visualization of affinity maps in the specific position.** The Location is marked by the red cross in each affinity map. TP represents our Tree Scanning Algorithm.

# D  More Qualitative Results

Fig. 6 displays additional qualitative comparisons between our algorithm and previous scanning strategies (*e.g.*, cross-scanning and raster-scanning), which shows our advanced capability to perceive detailed structural information and capture long-range dependencies.

# E  Statistical Significance

| Method | PIQA | Arc-Easy | SST | WinoGrande | LAM-ppl | Race | Openbookqa |
|---|---|---|---|---|---|---|---|
| MambaTreeL (Ours) | 0.011 | 0.010 | 0.016 | 0.014 | 0.553 | 0.014 | 0.018 |

Table 9: **Standard error on language model benchmarks.** LAM-ppl indicates LAMBADA [49].

We calculate the standard deviation of our MambaTreeL on language model benchmarks in the open-sourced lm-evaluation-harness project as shown in Table 9.

