# OpenReview forum: "MambaTree: Tree Topology is All You Need in State Space Model"
_NeurIPS.cc/2024/Conference — NeurIPS 2024 spotlight_

### Official Review · Reviewer_LPAH · 2024-07-04

**Soundness:** 4
**Presentation:** 4
**Contribution:** 4
**Rating:** 7
**Confidence:** 4

**Summary:**

This paper integrates tree structures into SSMs, enabling hierarchical data processing instead of the traditional sequential approach. It identifies a minimum spanning tree that considers locality, utilizing dynamic programming to prune edges from the grid graph. By adopting this tree structure, the length of the original sequence is reduced, allowing SSMs to handle long-range dependencies more effectively. As a result, the proposed GrootVL outperforms previous SSMs in image classification, detection, segmentation, and language understanding.

**Strengths:**

- Integrating hierarchical structures into sequential models is an important yet underexplored direction.
- This paper proposes an intuitive yet novel algorithm that implements tree scanning into SSM architectures.
- Enhancing long-range dependencies is a significant benefit of the tree structure. In theory, it can reduce the context length from $L$ to $\log L$.
- The experiments are extensive, demonstrating results in various image and language tasks.
- The paper is well-structured, featuring a clear logical flow and ablation studies.

**Weaknesses:**

1. Analysis on the learned structures

While the paper primarily highlights performance benefits, tree structures offer additional advantages, such as interpretability. For example, as illustrated in Figure 1, one can identify syntax structures in sentences or scene structures in images. Examining the learned tree structures for individual data in depth would be highly insightful.

---
2. Consistency of tree structures across layers

If I understand correctly, the current architecture constructs a tree for each block independently. Do the layers generate consistent trees? If not, how can one justify the validity of the discovered structure? Would it be helpful to regularize the tree structures to ensure consistency across layers?

---
3. Discussion of previous hierarchical structure approaches

The related work section lacks a comparison with previous hierarchical structure approaches. Classic recursive neural networks [1] pioneered learning hierarchical structures from images and texts, and this concept has been extended to modern Transformer architectures [2-4]. However, those approaches build trees in a bottom-up manner by gradually grouping the original tokens, while this paper proposes a top-down approach that prunes edges from grid graphs. This methodological difference is noteworthy and merits discussion. Additionally, other approaches to encoding trees into sequential architectures, such as [5], are also worth discussing.

[1] Socher et al. Parsing Natural Scenes and Natural Language with Recursive Neural Networks. ICML 2011.\
[2] Wang et al. Tree Transformer: Integrating Tree Structures into Self-Attention. EMNLP 2019.\
[3] Bolya et al. Token Merging: Your ViT But Faster. ICLR 2023.\
[4] Ke et al. Learning Hierarchical Image Segmentation For Recognition and By Recognition. ICLR 2024.\
[5] Shen et al. Ordered Neurons: Integrating Tree Structures into Recurrent Neural Networks. ICLR 2019.

**Questions:**

N/A

**Limitations:**

The paper clearly states its limitation: the current architecture is not implemented in a hardware-efficient manner. However, I believe the paper provides enough academic insights, and practical extensions could be left for future work.

---

> ### Author Rebuttal · Authors · 2024-08-07
>
> **Q1: Analysis of the learned structures.**
>
> **Ans:** Thanks for the valuable suggestions! We acknowledge that input-aware tree structures provide significant interpretability benefits, including preserving the intricate structural details in the vision content and enhancing the long-range modeling in a sequence. In addition, tree scanning algorithm captures the higher-order relationships between each unit, significantly expanding the feature space of the model compared to the second-order relationships handled by attention mechanisms. We will revise our paper by providing more analysis.
>
> **Q2: Consistency of tree structures across layers.**
>
> **Ans:** In our manuscript, we dynamically generate the tree topology based on the specific input feature for each block separately, which achieves higher performance. We have experimented to explore the effect of a regularized tree structure as noted by the reviewer. The results are shown in Table 11. GrootV-T* refers to each stage sharing the same tree structure. This approach enhances efficiency with only a minimal compromise in accuracy.
>
> **Tabel 11. Ablation study about tree scanning algorithm.**
> | **Method (224x224)**   | **Throughput (img/s)** | **Acc. (Top1)** |
> |--------------------|:-:|:-:|
> | Baseline (w/o TSA) | 373                | 82.60       |
> | GrootV-T           | 281                | 83.44       |
> | GrootV-T*          | 392                | 83.41       |
> We will update this ablation study in the revision.
>
> **Q3: Discussion of previous hierarchical structure approaches.**
>
> **Ans:** Thanks very much for the insightful perspective and suggestions. It is the main difference from the previous bottom-up hierarchical structure that our tree-based topology retains original vertices to propagate features in a top-down manner. Method[1] as mentioned by the reviewer, captures the hierarchical representation by encoding an ordered tree structure to the sequence. Compared to it, our tree topology is essentially an undirected and acyclic graph, which can be dynamically constructed based on the input signal. In a word, it’s an interesting topic. We will supplement the related work section in our revised manuscript.
>
> [1] Ordered Neurons: Integrating Tree Structures into Recurrent Neural Networks.

---

> > ### Comment · Reviewer_LPAH · 2024-08-10
> > **Response to the Rebuttal**
> >
> > Thank you for the rebuttal. I find this paper interesting, and since all other reviewers have anonymously recommended acceptance, I stand by my original evaluation.
> >
> > I'm looking forward to seeing the analysis of the learned structures (Q1) in the revised paper. It's also exciting to see that the consistency of tree structures across layers significantly improves throughput while maintaining accuracy (Q2). This makes sense, as the structure of data should generally remain consistent across layers unless there's a clear reason for it to change.

---

### Official Review · Reviewer_g9ag · 2024-07-05

**Soundness:** 3
**Presentation:** 3
**Contribution:** 3
**Rating:** 6
**Confidence:** 4

**Summary:**

This paper proposes a tree state space model (SSM) to perform feature propagation on an input-aware topology. The author explores tree topology in SSM from both vision and language sides, leading to GrootV and GrootL respectively. The proposed method exhibits strong empirical performances on mainstream tasks. Extensive ablations are conducted to verify the approach's effectiveness. Missing training efficiency and inference throughput.

**Strengths:**

Strength：

1. The motivation is clear and tree topology in building the scanning mechanism is novel to me.
2. Extensive experiments demonstrate the strong empirical performances of the proposed method.
3. Abundant ablation studies are conducted.

**Weaknesses:**

Weakness:

1. Missing efficiency performance. Mamba is known for its linear complexity and efficiency. However, there is no efficiency experiments to report this property of Groot-V/L. It's important for me to know the training efficiency and inference throughput especially for Groot-V.

**Questions:**

Please see the comments above.

**Limitations:**

Please see the comments above.

---

> ### Author Rebuttal · Authors · 2024-08-07
>
> **Q1: Missing efficiency performance.**
>
> **Ans:** Thanks for the suggestion. For inference throughputs, please refer to the "All Reviewer" section at the top of this rebuttal page. Besides, we provide the training throughputs of our method in Table 10, which are measured on a Nvidia V100 GPU with an input image scale of $224\times224$. We believe the efficiency can be further improved with more sophisticated optimization.
>
> **Table 10. Comparison of the throughputs during training.**
> || GrootV-T | GrootV-S | GrootV-B |
> |:-:|:-:|:-:|:-:|
> | Throughput (img/s) | 171      | 101      | 92       |
> We will update it in the revision!

---

### Official Review · Reviewer_xmLw · 2024-07-14

**Soundness:** 3
**Presentation:** 3
**Contribution:** 3
**Rating:** 7
**Confidence:** 4

**Summary:**

The paper introduces a tree scanning algorithm for state space models specifically for Mamba. The naive and fixed scan patterns like raster or local scans commonly used for vision tasks do not consider the topological structure of 2D image input. The proposed algorithm generates a minimum spanning tree which can help Mamba to model the semantic information of the input. The paper further introduces a dynamic programming procedure to avoid the quadratic complexity of the tree scanning algorithm. The experiments on various vision and language modeling tasks show that the tree topology scan helps improve accuracy.

**Strengths:**

1. The motivation and the proposed scan algorithm make sense and are easy to understand.
2. The proposed method can be useful not only for SSMs but also for other models that require sequential scans.
3. As the proposed scan already generates a structure of the input based on the relevance of tokens, the sequence learning process can potentially be minimal.
4. Using the proposed method, GrootV outperforms many recent baselines for multiple tasks (image classification/segmentation, object detection, and language understanding).

**Weaknesses:**

1. Regarding the root setting, the authors show that the root setting to all vertices outperforms the ones with only the first or the last. However, it increases the traverse time (1 vertex vs all vertices), and the accuracy improvement using all vertex is marginal (only 0.4% on Imagenet-1K). I understand that the dynamic programming procedure improves the speed but still increases by the sequence length $L$. I am still not fully convinced how effective this is. Can authors comment on it?

3. Related to point 2, there are no speed comparisons, especially with different scanning strategies, root settings, and the use of the dynamic programming procedure in practice.

1. Some details are missing in the paper.
    1. Figure 1 is not explained in the paper. What exactly is the parameter generator?  Is it the initialization stage of the state, input, and output matrices (A, B, C, and D), or is the projection of these parameters the same as in Mamba? In the tree state space model part, is the main difference using TSA instead of eq 4 compared to the original Mamba?
    2. In Figure 4, how is the specific position defined? Is it based on the root vertex setting?
    3. The cross scan used in Table 4 is not described.

**Questions:**

1. In eq 5, Ω is the index set of all vertices. As $S(E_{ij})$ aggregates the connected vertices, some vertices will be visited many times during the recurrent scan in Mamba. It seems like redundant computations. Could authors either empirically or theoretically justify why this is needed?

2. In language understanding, the tree scanning algorithm is applied during finetuning. Intuitively, this will change the causality of tokens trained during pretraining. Is there any specific reason the algorithm is not used during pre-training?

3. How is Eq 5 (state aggregation process) derived from Eq 4 (SSM state computation)? Could authors provide more explanation about it?

4. The semantic segmentation results seem marginal compared to other tasks, and this is the only dense prediction task. Can this be due to aggregation instead of processing all pixels by Mamba?

**Limitations:**

The limitation is discussed and isreasonable.

---

> ### Author Rebuttal · Authors · 2024-08-07
>
> **Q1: The traverse time of all vertices using dynamic programming algorithm.**
>
> **Ans:** Thanks for the comments, but we believe there exists some misunderstanding. Given a sequence with the length of $L$ with an established corresponding minimum spanning tree, for the case of single-vertex setting, we treat it as the root of a tree and aggregate features from other vertices, which operate in $O(L)$ complexity. While for the all-vertices setting, a naive approach treats each vertex as a root separately, resulting in $O(L^2)$ complexity. In contrast, this paper proposes a dynamic programming algorithm where a random vertex is chosen as the root, features are aggregated from leaf vertices to the root, followed by propagation from the root to the leaves, achieving the same effect. Therefore, the complexity of GrootV for all nodes remains linear at $O(L)$. We will further clarify it in the revision.
>
> **Q2: There are no speed comparisons.**
>
> **Ans:** For the comparison of inference speed, please refer to "All Reviewer" section at the top of this rebuttal page. We will further clarify it in the revision.
>
> **Q3: Some details are missing.**
>
> **Ans:** Thanks for your valuable suggestions!
>
> 1) Figure 1 illustrates a comparison of different propagation strategies for vision tasks and language tasks as discussed in Line 33 and Line 44 of the main manuscript.
>
> 2) The parameters generator in Figure 2 utilizes the same projection network as Mamba to unleash the context-aware capability of state space modeling. The only difference between Tree SSM and Mamba lies in the replacement of the structured state space block with the proposed tree scanning algorithm (referring to Line 130 of the main manuscript).
>
> 3) The anchor points shown in Figure 4 are randomly selected. We visualize the affinity maps of different positions to illustrate the capability of TSA to preserve detailed structural information. More qualitative results are shown in Sec. D of Appendix in the manuscript. Benefiting from the superiority of TSA, all pixels have equal access to long-range context.
>
> 4) The cross scan in Table 4 is the 4-directional raster-scanning strategy shown on the left side of Figure 1 in the manuscript.
>
> We will revise our paper to provide more details.
>
> **Q4: Explanation for why some vertices will be visited many times.**
>
> **Ans:** There could be some misunderstandings. For causal reasoning in text tasks, each node is visited only once. In image tasks, due to the use of the proposed dynamic programming algorithm, each node is only visited twice. The detailed mechanism refers to Section 3.2 of our manuscript and the answer in Q1.
>
> **Q5: Is there any specific reason the algorithm is not used during pre-training?**
>
> **Ans:** The core objective of this paper is to introduce a new structure for state space models. Through strict ablation studies, we validate the effectiveness of the method using LoRA fine-tuning for language tasks. Scaling up the models and exploring pre-training settings are left for future work.
>
> **Q6: How is Eq 5 derived from Eq 4?**
>
> **Ans:** The original Mamba state transition formula (Equation 4) can be easily derived into the form of Equation 5 as follows:
> $$
> \begin{aligned}
> h _ {i}&=\bar{\mathbf{A}} _ {i}h _ {i-1}+\bar{\mathbf{B}} _ {i}x _ {i}\\\\
>      &=\bar{\mathbf{A}} _ {i}\bar{\mathbf{A}} _ {i-1}h _ {i-2}+\bar{\mathbf{A}} _ {i}\bar{\mathbf{B}} _ {i-1} x _ {i-1}+\bar{\mathbf{B}} _ {i}x _ {i}\\\\
>      &\cdots\\\\
>      &={\textstyle\sum _ {j=1}^{i}}{\textstyle\prod _ {k=j+1}^{i}}\bar{\mathbf{A}} _ {k}\bar{\mathbf{B}} _ {j}x _ {j}
> \end{aligned}
> $$
> The feature aggregation described by the above formula is based on a linear topology (from the first vertex to $i$-$th$ vertex). If we propagate the state of each vertex along the built tree-topological path, Equation 4 is firstly transformed to the following formula (from children vertices to their parent vertex):
> $$
> h _ {i}={\textstyle\sum _ {j\in \\{ k|\text{par(k)=}i \\}}}\bar{\mathbf{A}} _ {j}h _ {j}+\bar{\mathbf{B}} _ {i}x _ {i}
> $$
> Then it can be easily derived into Equation 5 in a similar way as above, the detailed definition can be seen in Sec 3.2 of the manuscript.
>
> **Q7: The semantic segmentation results seem marginal compared to other tasks. Can this be due to aggregation instead of processing all pixels by Mamba?**
>
> **Ans:** As the same as Mamba-based methods, we aggregate features for all vertices, which is elaborated in Q1. Besides, as shown in Table 2 and Table 7 in the manuscript, our method shows consistent improvements over other SSM-based methods in terms of accuracy and efficiency.

---

> > ### Comment · Reviewer_xmLw · 2024-08-12
> > **Response to the rebuttal**
> >
> > Thank you for the answers.
> > My major concern was the efficiency of the algorithm. The answer to the speed comparison and the algorithm's details helped me understand the algorithm and resolved my concerns. Please make sure to add these details to the updated version.
> > The paper includes clear strengths and improves SSMs for vision and language models. I increase my rating.

---

### Official Review · Reviewer_NQEG · 2024-07-15

**Soundness:** 3
**Presentation:** 3
**Contribution:** 3
**Rating:** 5
**Confidence:** 4

**Summary:**

This paper studies the optimization of selective stat space modeling by particularly proposing the GrootVL model. Specifically, it firstly constructs the tree topology based on spatial information and then aggregates the features to enhance the representation informativeness. The proposed methods are versatile for both visual and textual tasks. The experimental results generally demonstrate the effectiveness of the proposed model.

**Strengths:**

1. This paper studies the problem of mamba framework optimization with a particular effort in the design tree structure learning module, which is an interesting investigation for the related domain.
2. The paper is written with good clarity and thus it is easy to follow.
3. The experimental results generally demonstrate the effectiveness of the proposed methods to support the claims made by this paper.

**Weaknesses:**

1. One of the major concerns is the insufficient efficiency analysis of the proposed Tree Scanning Algorithm in terms of complexity analysis and/or runtime costs for this related module.
2. Another concern is there is no significance test for the proposed model evaluation, especially when the metrics achieved by Grootvl are numerically close to the baselines. Furthermore, it is unclear the performance deviations in evaluation, which is also important to indicate the stability of the proposed model.

**Questions:**

1. Since the Tree structure is a graph as well, how to relate and/or differentiate the connections between graph-based mamba models, e.g., [1].
2. How many times of evaluations were performed to achieve the results?

[1] Graph-Mamba: Towards Long-Range Graph Sequence Modeling with Selective State Spaces

**Limitations:**

Please see the weakness for details.

---

> ### Author Rebuttal · Authors · 2024-08-07
>
> **Q1: More efficiency analysis.**
>
> **Ans:** Thanks for your valuable suggestions! We have introduced the complexity and optimized version of our method in Section 3.1 and Section B of our manuscript. For the comparison of inference time, please refer to "All Reviewer" section at the top of this rebuttal page. We will further clarify it in the revision.
>
> **Q2: There is no significance test and unclear illustrations on the performance deviations.**
>
> **Ans:** We adhere strictly to the same benchmark evaluation protocols to ensure a fair and standardized comparison with previous models. For language tasks, we have included comprehensive significance results in the Appendix of the manuscript (Section E). For vision tasks, as the reviewer mentioned, we have retrained our models three times for the segmentation task, whose standard variation is about 0.11%. The results demonstrate that our approach consistently improves performance compared to other counterparts. We will add more significance tests in the revision.
>
> **Q3: How to relate and/or differentiate between the Graph-based mamba models?**
>
> **Ans:** This is an insightful question. The primary difference between our GrootVL and Graph-Mamba[1] lies in the input type and topology construction manner. Graph-Mamba directly utilizes a graph structure as input, which includes both node and edge embeddings, and keeps the topology through the whole process. In contrast, GrootVL takes images or text as input and dynamically constructs the topology structure based on input feature.
>
> **Q4: How many times of evaluations were performed?**
>
> **Ans:** For language tasks, the evaluation is calculated three times by using the most popular benchmark, lm-evaluation-harness[2]. For vision tasks, we have additionally retrained Groot V-T three times on semantic segmentation and got 0.11% standard variation. We will add more clarification in the revision.
>
> [1] Graph-Mamba: Towards Long-Range Graph Sequence Modeling with Selective State Spaces
>
> [2] A framework for few-shot language model evaluation

---

### Author Rebuttal · Authors · 2024-08-07

**To All Reviewers (Reply about efficiency performance):**

We sincerely appreciate all reviewers and ACs for their precious time and valuable feedback. Given that reviewers NQEG, xmLw, and g9ag have raised concerns regarding efficiency comparison, we will respond to this issue in this section. All the source code will be made public.

**Q1: Inference speed comparison.**

**Ans:** As shown in Table 9, we report the inference throughputs of our method on a Nvidia V100 GPU. The GrootV-T* refers to each stage sharing the same tree topology structure, which enhances efficiency without compromising accuracy. To achieve better practical inference speed, we also introduce a cuda implementation optimized for GPUs. Compared with other counterparts, our approach exhibits superior effectiveness and faster inference speed. We will add this table to the revision and release the optimized cuda code.

**Table 9: Runtime comparison on a Nvidia V100 GPU during inference.**
|**Method(224x224)**|**Throughput (img/s)**|**GPU Memory**|**FLOPs**|**#Params.**|**Acc. (Top1)**|
|-|:-:|:-:|:-:|:-:|:-:|
| PlainMamba-L2      | 363             | 4204M               | 8.1G      | 25M          | 81.6     |
| VMamba-T           | 374             | 8646M               | 4.9G      | 31M          | 82.5     |
| LocalVMamba-T      | 311             | 11298M              | 5.7G      | 26M          | 82.7     |
| GrootV-T(one root) | 283             | 6012M               | 4.8G      | 30M          | 83.0     |
| GrootV-T           | 281             | 6471M               | 4.8G      | 30M          | 83.4     |
| GrootV-T*          | **392**         | 4800M               | 4.8G      | 30M          | **83.4** |

---

### Decision · Program_Chairs · 2024-09-25

**Decision:**

Accept (spotlight)

**Comment:**

The authors proposed GrootVL -- a tree-structured state-space model (SSM) which efficiently models long-range interactions across a range of tasks including image classification, semantic segmentation and language understanding.

Reviewers appreciated that the method can be applied to different SSMs, and achieves strong empirical results. As a result, they all recommended acceptance.

A common concern among reviewers was about the efficiency analysis. The table in the rebuttal answered this, so please include it in the camera-ready version of the paper. Different architectures have different efficiency characteristics, and hence it is important to report multiple efficiency indicators to give a fair representation (as also mentioned in more detail in https://arxiv.org/abs/2110.12894)